# Predictors of falls and fractures leading to hospitalisation in 36101 people with affective disorders: a large representative cohort study

Ruimin Ma ,[1] Gayan Perera ,[1] Eugenia Romano,[1] Davy Vancampfort,[2,3] Ai Koyanagi ,[4,5] Robert Stewart,[1,6] Christoph Mueller,[1,6] Brendon Stubbs[1,6]

RM and GP are joint first authors.

For numbered affiliations see end of article.

**Correspondence to**
Dr Ruimin Ma;
ruimin.1.ma@kcl.ac.uk

## ABSTRACT

**Objectives** To investigate predictors of falls and fractures leading to hospitalisation in people with affective disorders.

**Design** Cohort study.

**Setting** The South London and Maudsley National Health Service (NHS) Foundation Trust (SLaM) Biomedical Research Centre (BRC) Case Register.

**Participants** A large cohort of people with affective disorders (International Classification of Diseases- 10th version [ICD-10] codes F30–F34) diagnosed between January 2008 and March 2016 was assembled using data from the SLaM BRC Case Register.

**Primary and secondary outcome measures** Falls and fractures leading to hospitalisation were ascertained from linked national hospitalisation data. Multivariable Cox proportional hazards analyses were administrated to identify predictors of first falls and fractures.

**Results** Of 36101 people with affective disorders (mean age 44.4 years, 60.2% female), 816 (incidence rate 9.91 per 1000 person-years) and 1117 (incidence rate 11.92 per 1000 person-years) experienced either a fall or fracture, respectively. In multivariable analyses, older age, analgesic use, increased physical illness burden, previous hospital admission due to certain comorbid physical illnesses and increase in attendances to accident and emergency services following diagnosis were significant risk factors for both falls and fractures. Having a history of falls was a strong risk factor for recurrent falls, and a previous fracture was also associated with future fractures.

**Conclusions** Over a mean 5 years' follow-up, approximately 8% of people with affective disorders were hospitalised with a fall or fracture. Several similar factors were found to predict risk of falls and fracture, for example, older age, comorbid physical disorders and analgesic use. Routine screening for bone mineral density and fall prevention programmes should be considered for this clinical group.

## INTRODUCTION

Falls and fractures are strongly related to increased morbidity, mortality, disability and healthcare expenditure in the general population,[1] and they are a frequent issue in older adults.[2] They often overlap, with more than

## Strengths and limitations of this study

► Predictors of falls and fractures leading to hospitalisation in people with affective disorders were investigated with a large representative cohort.
► Data of the study were derived directly from the electronic health record.
► Falls and fractures leading to hospitalisation were ascertained from linked national hospitalisation data.
► We did not stratify according to different affective disorder subtypes and psychotropic medication categories.
► We had no information on lifestyle factors and type of fractures recorded at admission.

90% of hip fractures caused by falls,[3] and share similar and multifactorial causes, such as muscle weakness[4] and use of antiepileptic drugs.[5] Several government strategies have been implemented to prevent and assess falls in the general population.[6] Fractures, for example, hip and spine fractures increase the risk of future falls and fractures.[7] Over 3 million people in the UK are at a high risk of fractures,[8] and prevention has been acknowledged as a worldwide priority.[9] However, management is difficult, as causes for falls and fractures can be both intrinsic and extrinsic, and prevention requires both pharmacological and non-pharmacological interventions.[3]

Despite some progress in the general population, there is poor evidence on risk factors for falls and fractures for patients with affective disorders. People with mental health problems like depression and schizophrenia report higher risks of falls and fractures compared with the general population,[10 11] and despite poor evidence in people with bipolar disorders, a few studies suggest this population may also be at increased risk.[12 13]

Several studies have investigated the association between bone loss and depression, but there is a lack of large-scale cohort studies focusing on falls and fractures and related predictors. Another major limitation is that the majority of studies have relied on self-reported affective symptoms rather than clinical diagnosis.[14 15]

Given the increased concerns about the multifactorial causes of falls and fractures,[3–5] and as an answer to the lack of large-scale studies on the matter, this study examines a large representative cohort study to investigate predictors of falls and fractures leading to hospitalisation among people with clinically diagnosed affective disorders. In this way, we aim to provide results based on clinical diagnosis rather than self-reported data,[14 15] presenting a solid overview on potential risk factors for falls and fractures in the high-risk population of people with affective disorders.

## METHODS
A retrospective observational study was carried out using data from the South London and Maudsley National Health Service (NHS) Foundation Trust (SLaM) Biomedical Research Centre (BRC) Case Register. SLaM is one of the largest mental health and dementia care providers in Europe, serving a geographical catchment of four South London Boroughs (including Lambeth, Lewisham, Southwark and Croydon) with a population in excess of 1.3 million. Data for this study were retrieved using the Clinical Record Interactive Search (CRIS) platform, which enables a de-identified version of SLaM's electronic health record to be accessible for research projects within a robust and patient-led government framework.[16] The SLaM BRC Case Register has been described in detail elsewhere[17] and has supported a wide range of studies,[18 19] including longitudinal cohort studies investigating falls and fractures in other populations.[20 21] Data are currently archived in CRIS on more than 450 000 cases with a wide variety of mental disorders. Data from source-structured fields have been extensively supplemented through natural language process (NLP) applications using Generalised Architecture for Text Engineering software, applying information extraction techniques to derive structured information from the extensive text fields held in the mental health record.[17]

### Participants, study period and additional data sources
All SLaM patients with mood (affective) disorders (International Classification of Diseases - 10th version [ICD-10] codes F30–F34), diagnosed between 1 January 2008 and 3 March 2016, were included. Those patients with an age under 18 years at the time of mood disorders were excluded. SLaM patient records have been linked with national Hospital Episode Statistics (HES) which are compiled from all NHS Trusts in England (both acute and mental health services), including statistical abstracts of records of all inpatient episodes, as well as outpatient and emergency care.[17] In addition, CRIS data have been linked to the Office for National Statistics (ONS) mortality records over the same period.

### Patient and public involvement
The entire research programme was developed after extensive patient, public and carer input, who helped develop the study and contributed to the funding applicable. They co-developed the research questions, outcomes and contributed to factors included in the models. Moreover, they reinforced this research as a neglected topic of utmost importance to them. The patient and public involvement (PPI) group designed the study with the senior researcher of the paper (BS); they were involved in the conception of the idea, the planning of the study during bimonthly meetings to discuss study design, hypotheses, outcomes and what matters to patients of interest. They were also involved in helping interpret the data outputs and publication. Results will be summarised and made available to the PPI group through local newsletters and inform clinical services and local co-developed Recovery College Courses.

### Co-primary outcome: falls and any fractures
The co-primary outcomes were hospital admissions relating to a fall or any fracture extracted from linked HES data and all discharge diagnoses (primary or any secondary diagnosis codes) recorded between January 2008 and March 2016, and based on the following ICD-10 codes: (1) falls (W00–W19); (2) fractures (M80–M84, M907, S02, S12, S32, S42, S52, S62, S72, S82, S92, S22, T02, T08, T10, T12X, T902, T911, T912, T921). In addition, linked mortality records from the ONS linkage were examined for any instance of fall or fracture ICD codes in any cause of death field on the death certificate to identify the date of death attributed to a fall or fracture.

### Measurements
Several additional measurements were obtained from CRIS. All independent variables (covariates) were defined according to the value closest to the date of the first recorded mood (affective) disorder diagnosis. Demographic covariates comprised: age at diagnosis, gender and ethnicity (multiple codes categorised into: (1) white, consists of white British, white Irish and other white; and (2) non-white). Index of Multiple Deprivation (IMD 2015) for the neighbourhood of residence (ie, Lower Super Output Area, which consists of approximately 650 households) at the time of diagnosis was divided into quintiles according to deprivation scores with equal size allocated to each group. The IMD has previously been used in CRIS[22] and combines census-derived data at area level across several domains including income, employment, health, education, barriers to housing and services, living environment and crime.[23] Information on cohabiting status (cohabiting: married/civil partner, married, cohabiting; non-cohabiting: single, divorced, civil partnership dissolved, widowed, separated) was also ascertained at the index diagnosis.

## Illness burden

The Health of the Nation Outcome Scales (HoNOS)[24] are routinely administered measures of illness burden in UK mental health services and are recorded in structured fields on the electronic health record. Individual HoNOS item scores (agitated behaviour, self-injury, problem drinking and drugs, cognitive problems, physical illness, hallucinations, depressed mood, relationship problems, daily living problems, living condition problems, occupational problems) and dates were obtained within 6 months before or after the date of the index diagnosis, and the closest scores in time to this date were included in analyses. Each HoNOS item is rated on a Likert scale, ranging from 0 (ie, no problem) to 4 (ie, severe or very severe problem). A detailed glossary for HoNOS is reported elsewhere.[25] Scores 2 or over in the individual HoNOS were classified as having a problem on each item, generating binary covariates.

## Mental disorder comorbidity

Diagnoses of F00–F03 (dementia); F20–F29 (schizophrenia spectrum disorder); F40–F48 (neurotic, stress-related and somatoform disorders); F50 (eating disorders); and F60–F69 (disorders of adult personality and behaviour) were ascertained within 1 year before or after the index diagnosis of mood (affective) disorder.

## Medication

Medications received were extracted from structured medication fields in the record, supplemented by an NLP application applied to text fields ascertaining mentions of current medication.[17] Presence or not of the following medication groups was ascertained on the basis of information within 6 months before or after the index diagnosis: anticholinergics, antihypertensives, antidepressants, antipsychotics, anxiolytics and hypnotics, and analgesics. The total number of medications prescribed for any condition was calculated for each participant and used as a continuous variable.

## Physical comorbidity

Information on physical comorbidities was ascertained using the data linkage between CRIS and national HES records. Information on all ICD-10 diagnoses at discharge (primary or any secondary diagnosis codes) was ascertained from any hospitalisation within 6 months before or after the index diagnosis date and the following binary variables generated: (1) ischaemic heart disease (I20, I21, I22, including coronary heart disease (I25)); (2) arrhythmia (I44–I49, including atrial fibrillation (I48)); (3) heart failure (I50); (4) diabetes (E08, E09, E10, E11, E12, E13); (5) hypotension (I95–99); (6) hypercholesterolaemia (E78); (7) hypertension (I10–15); (8) urinary tract infections (UTIs) (N39); (9) osteoporosis (M80–85); (10) visual disturbance and blindness (H53–54); (11) hearing loss (H90–95); (12) syncope or collapse (R50–R69); (13) Parkinson's disease (G20). Furthermore, occurrence of falls and/or fractures before the

index diagnosis was also collected. Finally, the number of attendances to accident and emergency services (A&E) following the index diagnosis was also recorded.

## Statistical analysis

The study sample was described initially in terms of demographic and clinical variables, followed by unadjusted Cox proportional hazards models to predict the first fall and first fracture separately after the index diagnosis of mood (affective) disorder. The predictor variables at baseline used in the first univariate models included sociodemographic information (year of index diagnosis, mean age, gender, ethnicity, marital status), medications (antipsychotics, anxiolytics and hypnotics, antidepressants, analgesics, anticholinergics, antihypertensives), comorbid psychiatric diagnosis (F20–29 schizophrenia spectrum disorder, F40–48 neurotic/stress disorders, F00–F03 dementia, F50 eating disorders, F60 disorders of adult personality and behaviour), HoNOS scores (mean total and each individual item) and physical health comorbidities (as indicated above). Factors that yielded a p value of <0.10 in the univariate model for fall or fracture outcome were subsequently entered into the multivariable model. A final multivariable Cox proportional hazards model, using stepwise backward elimination technique where those variables not significant (p>0.05) were eliminated, with hazard ratios (HRs) and 95% confidence intervals (CIs) displayed. Having checked a correlation matrix of coefficients in the Cox model, the total number of medications received was substantially collinear with individual types of medication received; therefore, this total number was removed as a covariate. All analyses were conducted using STATA, V.13.

## RESULTS

The sample comprised 36101 people with a diagnosis of mood (affective) disorder (F30–F34) (mean age at first diagnosis 44.4 years, SD: 17.8, 60.2% female). Over a mean 5 years' follow-up, 2948 patients had a fall and/or a fracture recorded: 816 only a fall, 1117 only a fracture and 1015 both; 1831 with any fall and 2193 with any fracture (mean age at first fall 64.2 years, SD: 24.8; mean age at first fracture 62.5 years, SD: 22.8) (online supplemental table 1). The incidence rate of falls was 9.91 per 1000 person-years and that for fractures 11.92 per 1000 person-years. Table 1 summarises characteristics of those who had a recorded fall or fracture compared with those who did not.

## Length of hospital stay

The mean length of hospitalisation following a fall (n = 1831) was 7.9 days (range 0–374), for a total of 20767 full days in hospital. The mean length of stay in hospital following a fracture (n=2193) was 13.2 days (range 0–374), for a total of 40548 days. This equates to 18.51 years of inpatient hospital stay for 1000 person-years of follow-up due to a fall and 36.15 years of inpatient

**Table 1** Characteristics of those patients with mood (affective) disorders who were admitted to hospital with a fall/fracture after a diagnosis of mood (affective) disorders (F30–F34)

| Characteristics of sample | Presence of falls | | Presence of fractures | |
|---|---|---|---|---|
| | No (n=34 270) | Yes (n=1831) | No (n=33 908) | Yes (n=2193) |
| Age at the time of mood (affective) disorder diagnosis | | | | |
| 18–34 | 12 685 (37.0) | 205 (11.2) | 12 572 (37.1) | 318 (14.5) |
| 35–49 | 11 409 (33.3) | 336 (18.4) | 11 346 (33.5) | 399 (18.2) |
| 50–64 | 5829 (17.0) | 362 (19.8) | 5751 (17.0) | 440 (20.1) |
| 65–79 | 2797 (8.2) | 485 (26.5) | 2728 (8.0) | 554 (25.3) |
| 80 and over | 1550 (4.5) | 443 (24.2) | 1511 (4.5) | 482 (22.0) |
| Gender | | | | |
| Female | 20 627 (60.2) | 1107 (60.5) | 20 348 (60.0) | 1386 (63.2) |
| Male | 13 638 (39.8) | 724 (39.5) | 13 555 (40.0) | 807 (36.8) |
| Ethnicity | | | | |
| White | 20 772 (60.6) | 1519 (83.0) | 20 536 (60.6) | 1755 (80.0) |
| Non-white | 12 450 (36.3) | 288 (15.7) | 12 333 (36.4) | 405 (18.5) |
| Marital status | | | | |
| Cohabiting | 7957 (23.2) | 416 (22.7) | 7867 (23.2) | 506 (23.1) |
| Non-cohabiting | 23 028 (67.2) | 1345 (73.5) | 22 776 (67.2) | 1597 (72.8) |
| Index of Multiple Deprivation 2015 (SD) | 27.97 (11.31) | 27.90 (11.53) | 27.97 (11.30) | 27.97 (11.52) |
| Least deprived quintile | 6657 (19.4) | 388 (21.2) | 6593 (19.4) | 452 (20.6) |
| 2nd most deprived quintile | 6682 (19.5) | 378 (20.6) | 6643 (19.6) | 417 (19.0) |
| 3rd most deprived quintile | 6706 (19.6) | 354 (19.3) | 6637 (19.6) | 423 (19.3) |
| Most deprived quintile | 6708 (19.6) | 346 (18.9) | 6612 (19.5) | 442 (20.2) |
| Medication prescription (within 6 months before or after mood (affective) disorder diagnosis) | | | | |
| Anticholinergics | 17 714 (51.7) | 1089 (59.5) | 17 489 (51.6) | 1314 (59.9) |
| Antihypertensives | 2561 (7.5) | 386 (21.1) | 2553 (7.5) | 394 (18.0) |
| Antidepressants | 19 351 (56.5) | 1168 (63.8) | 19 113 (56.4) | 1406 (64.1) |
| Antipsychotics | 8777 (25.6) | 469 (25.6) | 8694 (25.6) | 552 (25.2) |
| Anxiolytics and hypnotics | 9443 (27.6) | 558 (30.5) | 9349 (27.6) | 652 (29.7) |
| Analgesics | 3083 (9.0) | 405 (22.1) | 3059 (9.0) | 429 (19.6) |
| Number of medications received (within 6 months before or after mood (affective) disorder diagnosis) | | | | |
| 0 | 10 207 (29.8) | 394 (21.5) | 10 126 (29.9) | 475 (21.7) |
| 1 | 5330 (15.6) | 259 (14.1) | 5275 (15.6) | 314 (14.3) |
| 2 | 7062 (20.6) | 360 (19.7) | 6961 (20.5) | 461 (21.0) |
| 3 | 6599 (19.3) | 361 (19.7) | 6503 (19.2) | 457 (20.8) |
| 4 | 3858 (11.3) | 304 (16.6) | 3839 (11.3) | 323 (14.7) |
| 5 | 1038 (3.0) | 121 (6.6) | 1029 (3.0) | 130 (5.9) |
| 6 | 176 (0.5) | 32 (1.7) | 175 (0.5) | 33 (1.5) |
| Other psychiatric conditions (within 6 months before or after mood (affective) disorder diagnosis) | | | | |
| F00–F03 (dementia) | 754 (2.2) | 207 (11.3) | 749 (2.2) | 212 (9.7) |
| F20–F29 (schizophrenia spectrum disorder) | 2701 (7.9) | 119 (6.5) | 2676 (7.9) | 144 (6.6) |
| F30–F31 (bipolar affective disorder) | 5489 (16.0) | 222 (12.1) | 5475 (16.1) | 236 (10.8) |
| F50 (eating disorders) | 447 (1.3) | 13 (0.7) | 441 (1.3) | 19 (0.9) |
| F40–F48 (neurotic, stress-related and somatoform disorders) | 3652 (10.7) | 147 (8.0) | 3601 (10.6) | 198 (9.0) |
| F60 (disorders of adult personality and behaviour) | 1436 (4.2) | 77 (4.2) | 1428 (4.2) | 85 (3.9) |
| Problem HoNOS (score 2 or over) (within 6 months before or after mood (affective) disorder diagnosis) | | | | |

Continued

**Table 1** Continued

| Characteristics of sample | Presence of falls | | Presence of fractures | |
|---|---|---|---|---|
| | No (n=34 270) | Yes (n=1831) | No (n=33 908) | Yes (n=2193) |
| Agitated behaviour | 3571 (10.4) | 234 (12.8) | 3547 (10.5) | 258 (11.8) |
| Self-injury | 3586 (10.5) | 186 (10.2) | 3545 (10.5) | 227 (10.4) |
| Problem drinking drugs | 2957 (8.6) | 204 (11.1) | 2932 (8.6) | 229 (10.4) |
| Cognitive problems | 2821 (8.2) | 349 (19.1) | 2792 (8.2) | 378 (17.2) |
| Physical illness | 6514 (19.0) | 770 (42.1) | 6355 (18.7) | 929 (42.4) |
| Hallucinations | 3198 (9.3) | 175 (9.6) | 3176 (9.4) | 197 (9.0) |
| Depressed mood | 14 582 (42.6) | 766 (41.8) | 14 390 (42.4) | 958 (43.7) |
| Relationship problems | 7644 (22.3) | 342 (18.7) | 7560 (22.3) | 426 (19.4) |
| Daily living problems | 5854 (17.1) | 576 (31.5) | 5723 (16.9) | 707 (32.2) |
| Living condition problems | 3431 (10) | 190 (10.4) | 3404 (10.0) | 217 (9.9) |
| Occupational problems | 5574 (16.3) | 392 (21.4) | 5513 (16.3) | 453 (20.7) |
| Mean overall HoNOS score (SD) | 10.67 (5.63) | 11.64 (5.53) | 10.64 (5.63) | 11.91 (5.46) |
| Number with missing HoNOS | 12 545 (36.6) | 523 (28.6) | 12 411 (36.6) | 657 (30.0) |
| Hospital admissions (within 6 months before or after mood (affective) disorder diagnosis) | | | | |
| Ischaemia+CHD+IHD | 1128 (3.3) | 210 (11.5) | 1089 (3.2) | 249 (11.4) |
| Arrhythmia+AF | 1003 (2.9) | 197 (10.8) | 967 (2.9) | 233 (10.6) |
| Heart failure | 437 (1.3) | 88 (4.8) | 417 (1.2) | 108 (4.9) |
| Diabetes | 1516 (4.4) | 264 (14.4) | 1488 (4.4) | 292 (13.3) |
| Hypotension | 435 (1.3) | 104 (5.7) | 424 (1.3) | 115 (5.2) |
| Hypercholesterolaemia | 1097 (3.2) | 221 (12.1) | 1090 (3.2) | 228 (10.4) |
| Hypertension | 2837 (8.3) | 546 (29.8) | 2752 (8.1) | 631 (28.8) |
| Urinary tract infections | 1366 (4.0) | 324 (17.7) | 1297 (3.8) | 393 (17.9) |
| Osteoporosis | 409 (1.2) | 121 (6.6) | 183 (0.5) | 347 (15.8) |
| Visual disturbance and blindness | 241 (0.7) | 53 (2.9) | 246 (0.7) | 48 (2.2) |
| Hearing loss | 189 (0.6) | 41 (2.2) | 174 (0.5) | 56 (2.6) |
| Syncope or collapse | 1990 (5.8) | 444 (24.2) | 2022 (6.0) | 412 (18.8) |
| Parkinson's disease | 116 (0.3) | 45 (2.5) | 128 (0.4) | 33 (1.5) |
| Falls before diagnosis | 750 (2.2) | 267 (14.6) | 749 (2.2) | 268 (12.2) |
| Fractures before diagnosis | 1075 (3.1) | 231 (12.6) | 860 (2.5) | 446 (20.3) |
| Mean number of attendances to A&E following mood (affective) disorder diagnosis (SD) | 3.94 (9.98) | 16.64 (28.88) | 4.00 (9.62) | 13.67 (29.35) |

A&E, accident and emergency services; AF, atrial fibrillation; CHD, coronary heart disease; HoNOS, Health of the Nation Outcome Scales; IHD, ischaemic heart disease.

hospital stay for 1000 person-years of follow-up due to a fracture. For the 1831 patients reporting a fall, the mean hospital admissions due to a fall was 1.5 (range 1–15); for the 2193 patients reporting a fracture, the mean number of hospital admissions due to a fracture was 2.3 (range 1–42).

## Factors associated with falls and fractures

Cox proportional hazards models analysing unadjusted predictors of falls and fractures (95% CI) are reported in table 2.

## Multivariable predictors of falls

Multivariable models of factors associated with first fall hospital admission are presented in table 3. In model 2, older age was strongly associated with higher risk and non-European ethnicity with lower risk, but neighbourhood deprivation reported no association. Analgesics had a significant association (with increased risk); and of comorbid psychiatric conditions, only the ICD-10 F40–F48 group had a significant association (with reduced risk). Of the HoNOS items, cognitive problems and physical illness were associated with higher risk, and depressed mood problems with lower risk. Higher risk of falls was associated with several hospitalisation discharge diagnoses, namely heart failure, diabetes, hypotension, UTI, osteoporosis, and syncope or collapse. Hospitalised fall was associated with having a history of falls reported before the index diagnosis and with increased A&E attendance after the index diagnosis.

**Table 2** Univariate Cox proportional hazards model (95% CI) showing factors affecting time to first fall/fracture hospital admission since diagnosis of mood (affective) disorders

| Characteristics | Outcome falls | | Outcome fractures | |
|---|---|---|---|---|
| | HR (95% CI) | P value | HR (95% CI) | P value |
| Age at the time of mood (affective) disorder diagnosis | | | | |
| 18–34 | Ref | | Ref | |
| 35–49 | 1.74 (1.46 to 2.07) | <0.001 | 1.33 (1.15 to 1.54) | <0.001 |
| 50–64 | 3.88 (3.27 to 4.61) | <0.001 | 3.03 (2.62 to 3.50) | <0.001 |
| 65–79 | 12.41 (10.54 to 14.62) | <0.001 | 9.11 (7.94 to 10.46) | <0.001 |
| 80 and over | 26.94 (22.8 to 31.83) | <0.001 | 18.46 (16.00 to 21.29) | <0.001 |
| Female gender | 0.99 (0.90 to 1.09) | 0.87 | 1.12 (1.03 to 1.22) | 0.01 |
| Non-European ethnicity | 0.32 (0.28 to 0.37) | <0.001 | 0.39 (0.35 to 0.44) | <0.001 |
| Non-cohabiting marital status | 1.08 (0.97 to 1.21) | 0.17 | 1.05 (0.95 to 1.16) | 0.31 |
| Deprivation quintile (IMD 2015) | | | | |
| Least deprived quintile | Ref | | Ref | |
| 2nd least deprived quintile | 0.83 (0.71 to 0.96) | 0.01 | 0.89 (0.78 to 0.99) | 0.04 |
| 3rd most deprived quintile | 0.88 (0.76 to 1.02) | 0.09 | 0.90 (0.79 to 1.03) | 0.13 |
| 2nd most deprived quintile | 0.94 (0.82 to 1.09) | 0.43 | 0.89 (0.78 to 1.00) | 0.05 |
| Most deprived quintile | 0.86 (0.74 to 0.99) | 0.04 | 0.94 (0.82 to 1.07) | 0.35 |
| Medication prescribed (within 1 year before or after mood (affective) disorder diagnosis) | | | | |
| Anticholinergics received | 1.42 (1.3 to 1.56) | <0.001 | 1.45 (1.33 to 1.57) | <0.001 |
| Antihypertensives received | 3.69 (3.3 to 4.13) | <0.001 | 2.98 (2.67 to 3.33) | <0.001 |
| Antidepressants received | 1.41 (1.29 to 1.56) | <0.001 | 1.43 (1.31 to 1.56) | <0.001 |
| Antipsychotics received | 0.95 (0.86 to 1.06) | 0.35 | 0.93 (0.84 to 1.02) | 0.12 |
| Anxiolytics and hypnotics received | 1.11 (1.00 to 1.22) | 0.05 | 1.06 (0.97 to 1.17) | 0.18 |
| Analgesics received | 2.97 (2.66 to 3.32) | <0.001 | 2.52 (2.27 to 2.80) | <0.001 |
| Increase in one type of polypharmacy | 1.21 (1.18 to 1.25) | <0.001 | 1.18 (1.15 to 1.21) | <0.001 |
| Presence of other psychiatric conditions (within 1 year before or after mood (affective) disorder diagnosis) | | | | |
| F00–F03 (dementia) | 6.90 (5.97 to 7.98) | <0.001 | 5.65 (4.90 to 6.51) | <0.001 |
| F20–F29 (schizophrenia spectrum disorder) | 0.75 (0.62 to 0.91) | <0.001 | 0.76 (0.64 to 0.90) | 0.01 |
| F30–F31 (bipolar affective disorder) | 0.71 (0.62 to 0.82) | <0.001 | 0.61 (0.54 to 0.71) | <0.001 |
| F40–F48 (neurotic to stress-related and somatoform disorders) | 0.76 (0.64 to 0.90) | <0.001 | 0.86 (0.75 to 0.99) | 0.05 |
| F50 (eating disorders) | 0.53 (0.31 to 0.91) | 0.02 | 0.64 (0.41 to 1.01) | 0.06 |
| F60 (disorders of adult personality and behaviour) | 0.95 (0.76 to 1.20) | 0.68 | 0.87 (0.7 to 1.09) | 0.22 |
| Problem in HoNOS (scored 2 or more) | | | | |
| Agitated behaviour | 1.11 (0.97 to 1.28) | 0.14 | 1.03 (0.90 to 1.17) | 0.71 |
| Self-injury | 0.84 (0.72 to 0.98) | 0.02 | 0.87 (0.76 to 1.01) | 0.06 |
| Problem drinking drugs | 1.11 (0.96 to 1.29) | 0.17 | 1.05 (0.91 to 1.21) | 0.48 |
| Cognitive problems | 2.66 (2.35 to 3.00) | <0.001 | 2.37 (2.11 to 2.66) | <0.001 |
| Physical illness | 3.88 (3.48 to 4.34) | <0.001 | 4.17 (3.77 to 4.63) | <0.001 |
| Hallucinations | 0.87 (0.74 to 1.02) | 0.08 | 0.82 (0.71 to 0.96) | 0.01 |
| Depressed mood | 0.75 (0.67 to 0.83) | <0.001 | 0.88 (0.80 to 0.98) | 0.02 |

**Table 2** Continued

| Characteristics | Outcome falls | | Outcome fractures | |
|---|---|---|---|---|
| | HR (95% CI) | P value | HR (95% CI) | P value |
| Relationship problems | 0.65 (0.58 to 0.74) | **<0.001** | 0.71 (0.63 to 0.79) | **<0.001** |
| Daily living problems | 2.41 (2.16 to 2.69) | **<0.001** | 2.61 (2.36 to 2.89) | **<0.001** |
| Living condition problems | 0.93 (0.80 to 1.08) | 0.34 | 0.89 (0.77 to 1.02) | 0.1 |
| Occupational problems | 1.30 (1.15 to 1.47) | **<0.001** | 1.30 (1.16 to 1.45) | **<0.001** |
| Overall increase in one unit of HoNOS | 1.04 (1.03 to 1.05) | **<0.001** | 1.04 (1.04 to 1.05) | **<0.001** |
| Admitted to general hospital (within 1 year before or after mood (affective) disorder diagnosis) | | | | |
| Ischaemia+CHD+IHD | 5.25 (4.54 to 6.06) | **<0.001** | 5.17 (4.53 to 5.90) | **<0.001** |
| Arrhythmia+AF | 6.30 (5.43 to 7.31) | **<0.001** | 6.10 (5.32 to 6.99) | **<0.001** |
| Heart failure | 7.81 (6.29 to 9.68) | **<0.001** | 7.89 (6.49 to 9.59) | **<0.001** |
| Diabetes | 4.66 (4.09 to 5.31) | **<0.001** | 4.18 (3.69 to 4.73) | **<0.001** |
| Hypotension | 7.29 (5.98 to 8.90) | **<0.001** | 6.51 (5.39 to 7.86) | **<0.001** |
| Hypercholesterolaemia | 5.14 (4.47 to 5.92) | **<0.001** | 4.24 (3.70 to 4.87) | **<0.001** |
| Hypertension | 6.08 (5.50 to 6.72) | **<0.001** | 5.78 (5.26 to 6.34) | **<0.001** |
| Urinary tract infections | 7.24 (6.42 to 8.17) | **<0.001** | 7.36 (6.59 to 8.21) | **<0.001** |
| Osteoporosis | 7.75 (6.44 to 9.32) | **<0.001** | 36.39 (32.35 to 40.94) | **<0.001** |
| Visual disturbance and blindness | 5.25 (3.99 to 6.90) | **<0.001** | 3.78 (2.84 to 5.03) | **<0.001** |
| Hearing loss | 5.57 (4.09 to 7.60) | **<0.001** | 6.45 (4.95 to 8.41) | **<0.001** |
| Syncope or collapse | 6.17 (5.54 to 6.86) | **<0.001** | 4.27 (3.84 to 4.76) | **<0.001** |
| Parkinson's disease | 9.83 (7.31 to 13.23) | **<0.001** | 5.60 (3.97 to 7.89) | **<0.001** |
| Falls before mood (affective) disorder diagnosis | 8.88 (7.8 to 10.12) | **<0.001** | 7.03 (6.18 to 7.99) | **<0.001** |
| Fractures before mood (affective) disorder diagnosis | 5.31 (4.62 to 6.10) | **<0.001** | 10.67 (9.61 to 11.85) | **<0.001** |
| Increase in one attendance to A&E following mood (affective) disorder diagnosis | 1.01 (1.01 to 1.01) | **<0.001** | 1.01 (1.01 to 1.01) | **<0.001** |

Factors with a p value < 0.10 are marked in bold
A&E, accident and emergency services; AF, atrial fibrillation; CHD, coronary heart disease; HoNOS, Health of the Nation Outcome Scales; IHD, ischaemic heart disease; IMD, Index of Multiple Deprivation.

## Multivariable predictors of fractures

Table 4 presents multivariable models of factors associated with first fracture hospitalisation. Fracture risk was increased in older patients and women and was decreased in those of non-European. Fracture risk was lower in patients receiving antihypertensives and higher in those receiving analgesics. It was also lower in those with neurotic, stress-related and somatoform disorders (ie, F40–F48), as well as bipolar affective disorder and manic episodes (ie, F30–F31). An elevated fracture risk was associated with physical illness on the HoNOS. Fracture risk was independently predicted by previous hospitalisation for heart failure, diabetes, UTI, osteoporosis, hearing loss and preceding fracture-related (but not fall-related) hospitalisation. Fracture risk was also associated with higher levels of post-diagnostic A&E attendance.

## DISCUSSION

To our knowledge, the current study is the first using representative data to investigate the predictors of hospitalised falls and fractures in people with clinically diagnosed affective disorders. Our data suggest that out of 36 101 people with affective disorders, 816 (ie, 2.26%) and 1117 (ie, 3.09%) of patients experienced a fall and fracture, respectively. Length of hospital stay was considerable, equating to 18.51 and 36.15 years of inpatient hospital stay for 1000 person-years of follow-up due to a fall and a fracture, respectively. The key factors increasing the risk of a hospitalised fall were older age, analgesic use, increased illness burden due to cognitive problems and physical illness, a history of general hospital admission due to comorbid physical illnesses (in particular syncope or collapse), increase in one attendance to A&E following

**Table 3** Two models showing predictors of first fall hospital admission among mood (affective) disorders (used stepwise removal of factors that were not significant at p=0.05)

| Characteristics | Falls | | | | |
| --- | --- | --- | --- | --- | --- |
| | Model 1 (n=20 938) | | Model 2 (n=22 414) | | |
| | HR (95% CI) | P value | HR (95% CI) | P value | |
| **Age at the time of mood (affective) disorder diagnosis** | | | | | |
| 18–34 | Ref | | Ref | | |
| 35–49 | 1.59 (1.23 to 2.06) | **<0.001** | 1.58 (1.24 to 2.02) | **<0.001** | |
| 50–64 | 3.11 (2.40 to 4.01) | **<0.001** | 2.98 (2.33 to 3.81) | **<0.001** | |
| 65–79 | 8.07 (6.31 to 10.33) | **<0.001** | 7.65 (6.07 to 9.64) | **<0.001** | |
| 80 and over | 12.40 (9.51 to 16.17) | **<0.001** | 11.80 (9.25 to 15.05) | **<0.001** | |
| Non-European ethnicity | 0.43 (0.36 to 0.51) | **<0.001** | 0.45 (0.38 to 0.52) | **<0.001** | |
| **Deprivation quintile (IMD 2015)** | | | | | |
| Least deprived quintile | Ref | | | | |
| 2nd least deprived quintile | 0.99 (0.83 to 1.19) | 0.92 | | | |
| 3rd most deprived quintile | 1.14 (0.95 to 1.36) | 0.17 | | | |
| 2nd most deprived quintile | 1.15 (0.96 to 1.38) | 0.13 | | | |
| Most deprived quintile | 1.00 (0.84 to 1.20) | 0.85 | | | |
| **Medication prescribed (within 1 year before or after diagnosis of mood (affective) disorders)** | | | | | |
| Anticholinergics received | 0.95 (0.74 to 1.22) | 0.68 | | | |
| Antihypertensives received | 0.98 (0.79 to 1.21) | 0.82 | | | |
| Antidepressants received | 0.91 (0.74 to 1.11) | 0.36 | | | |
| Anxiolytics and hypnotics received | 1.00 (0.80 to 1.24) | 0.97 | | | |
| Analgesics received | 1.36 (1.10 to 1.68) | **0.01** | 1.39 (1.22 to 1.58) | **<0.001** | |
| Increase in one type of polypharmacy | 1.02 (0.87 to 1.19) | 0.8 | | | |
| **Presence of other psychiatric conditions (within 1 year before or after diagnosis of mood (affective) disorders)** | | | | | |
| F00–F03 (dementia) | 1.05 (0.86 to 1.27) | 0.64 | | | |
| F20–F29 (schizophrenia spectrum disorder) | 1.07 (0.85 to 1.34) | 0.57 | | | |
| F30–F31 (bipolar affective disorder) | 0.87 (0.72 to 1.06) | 0.16 | | | |
| F40–F48 (neurotic to stress-related and somatoform disorders) | 0.74 (0.60 to 0.91) | **0.01** | 0.75 (0.61 to 0.91) | **0.01** | |
| F50 (eating disorders) | 0.86 (0.36 to 2.09) | 0.74 | | | |
| **One unit increase in HoNOS** | | | | | |
| Self-injury | 1.10 (0.92 to 1.32) | 0.3 | | | |
| Cognitive problems | 1.25 (1.06 to 1.46) | **0.01** | 1.28 (1.12 to 1.46) | **<0.001** | |
| Physical illness | 1.23 (1.07 to 1.42) | **<0.001** | 1.25 (1.10 to 1.42) | **<0.001** | |
| Depressed mood | 0.80 (0.70 to 0.92) | **<0.001** | 0.83 (0.74 to 0.93) | **<0.001** | |
| Relationship problems | 0.98 (0.84 to 1.14) | 0.77 | | | |
| Daily living problems | 1.01 (0.87 to 1.17) | 0.94 | | | |
| Occupational problems | 0.90 (0.78 to 1.03) | 0.13 | | | |
| **Admitted to general hospital (within 1 year before or after diagnosis of mood (affective) disorders)** | | | | | |
| Ischaemia+CHD+IHD | 0.95 (0.78 to 1.15) | 0.59 | | | |
| Arrhythmia+AF | 0.88 (0.73 to 1.07) | 0.2 | | | |
| Heart failure | 1.37 (1.05 to 1.78) | **0.02** | 1.27 (1.00 to 1.63) | **0.05** | |

Continued

**Table 3** Continued

| Characteristics | Falls | | | |
| | Model 1 (n=20 938) | | Model 2 (n=22 414) | |
| | HR (95% CI) | P value | HR (95% CI) | P value |
| --- | --- | --- | --- | --- |
| Diabetes | 1.37 (1.16 to 1.63) | **<0.001** | 1.36 (1.16 to 1.59) | **<0.001** |
| Hypotension | 1.38 (1.09 to 1.75) | **0.01** | 1.34 (1.07 to 1.68) | **0.01** |
| Hypercholesterolaemia | 1.16 (0.96 to 1.39) | 0.13 | | |
| Hypertension | 1.05 (0.88 to 1.23) | 0.63 | | |
| Urinary tract infections | 1.33 (1.13 to 1.56) | **<0.001** | 1.33 (1.14 to 1.54) | **<0.001** |
| Osteoporosis | 1.41 (1.08 to 1.83) | **0.01** | 1.37 (1.11 to 1.70) | **<0.001** |
| Visual disturbance and blindness | 1.22 (0.86 to 1.71) | 0.26 | | |
| Hearing loss | 0.91 (0.64 to 1.30) | 0.61 | | |
| Syncope or collapse | 1.94 (1.71 to 2.26) | **<0.001** | 2.05 (1.79 to 2.36) | **<0.001** |
| Parkinson's disease | 1.25 (0.89 to 1.77) | 0.2 | | |
| Falls before mood (affective) disorder diagnosis | 1.81 (1.51 to 2.18) | **<0.001** | 1.81 (1.54 to 2.13) | **<0.001** |
| Fractures before mood (affective) disorder diagnosis | 0.98 (0.78 to 1.22) | 0.86 | | |
| Increase in one attendance to A&E following mood (affective) disorder diagnosis | 1.01 (1.01 to 1.01) | **<0.001** | 1.01 (1.01 to 1.01) | **<0.001** |

Factors with a p value < 0.05 are marked in bold

A&E, accident and emergency services; AF, atrial fibrillation; CHD, coronary heart disease; HoNOS, Health of the Nation Outcome Scales; IHD, ischaemic heart disease; IMD, Index of Multiple Deprivation.

affective disorder diagnosis, and falls before the diagnosis. Similar risk factors for fractures were noted.

Approximately 8.2% of our sample experienced a hospital admission due to either a fall or fracture. Studies on healthy populations have suggested several mechanisms which may explain these elevated risks in our sample, such as vitamin D deficiency[26 27] and use of antidepressants.[28] Several studies also proposed a negative effect of leptin on bone mass and bone formation through a hypothalamic relay.[29] Some lifestyle factors seem to also be associated with a poor bone health, including physical inactivity[30] and alcohol consumption.[31] Previous studies have hypothesised other risk factors contributing to falls and fractures in people with depressive disorders,[32] but no large-scale study has yet focused on patients with clinically diagnosed affective disorders. Our data advance the current literature by providing detailed evidence on how a multitude of demographic, medical and psychological factors can differently affect risks of falls and fractures in people with affective disorder.

The association between older age and incidence of fall may be due to age-related physical conditions, as well as physiological (eg, loss of bone mineral density [BMD])[33] and pathological changes (eg, decreased cardiovascular functions).[34] Female gender is also a well-established risk factor for osteoporosis,[35] consistent with our finding. Surprisingly, we found no relationship between

deprivation and falls and fractures, although previous evidence has associated deprivation with fractures across different age groups.[36 37] We might suppose that while deprivation is associated with poor nutrition and physical inactivity, people with affective disorders have a poorer lifestyle, compared with the general population,[38] regardless of their financial status or living condition. Finally, the protective role of a non-European ethnic background is comparable with previous evidence supporting a higher fracture rate in Caucasian women compared with women from other ethnic groups,[39] and a lower fall rate in older immigrants than those with an English-speaking background.[40]

Our results provide new evidence on how hospitalisation due to falls or fractures can be an important burden for the NHS resources and for people with affective disorders, who are already at risk of prolonged hospitalisation.[41] Given the long hospital stay among our sample, its burden on NHS costs is unsurprising. The total cost of fracture is estimated to reach £4.4 billion per year in the UK alone.[8] Several items from the HoNOS were also associated with increased hazard of hospitalisation due to falls or fractures (eg, cognitive problems). Evidence on the role of these factors on hospital stay among this population is warranted for future research.

Our study reports stress-related and somatoform disorders to be associated with a decreased risk of falls and

**Table 4** Two models showing predictors of first fracture hospital admission among patients with mood (affective) disorders

| Characteristics | Model 1 (n=20 936) | | Model 2 (n=22 322) | |
|---|---|---|---|---|
| | HR (95% CI) | P value | HR (95% CI) | P value |
| Age at the time of mood (affective) disorder diagnosis | | | | |
| 18–34 | Ref | | Ref | |
| 35–49 | 1.27 (1.02 to 1.57) | **0.03** | 1.27 (1.03 to 1.55) | **0.02** |
| 50–64 | 2.31 (1.85 to 2.88) | **<0.001** | 2.28 (1.85 to 2.81) | **<0.001** |
| 65–79 | 5.75 (4.65 to 7.11) | **<0.001** | 5.55 (4.54 to 6.77) | **<0.001** |
| 80 and over | 7.46 (5.90 to 9.45) | **<0.001** | 7.37 (5.93 to 9.16) | **<0.001** |
| Female gender | 1.21 (1.08 to 1.36) | **<0.001** | 1.15 (1.03 to 1.29) | **0.01** |
| Non-European ethnicity | 0.59 (0.51 to 0.68) | **<0.001** | 0.59 (0.51 to 0.67) | **<0.001** |
| Deprivation quintile (IMD 2015) | | | | |
| Least deprived quintile | Ref | | | |
| 2nd least deprived quintile | 1.07 (0.91 to 1.28) | 0.41 | | |
| 3rd most deprived quintile | 1.11 (0.94 to 1.32) | 0.23 | | |
| 2nd most deprived quintile | 1.17 (0.99 to 1.39) | 0.07 | | |
| Most deprived quintile | 1.00 (0.85 to 1.19) | 0.96 | | |
| Medication prescribed (within 1 year before or after diagnosis of mood (affective) disorders) | | | | |
| Anticholinergics received | 1.08 (0.85 to 1.26) | 0.76 | | |
| Antihypertensives received | 0.76 (0.65 to 0.91) | **<0.001** | 0.80 (0.70 to 0.92) | **<0.001** |
| Antidepressants received | 1.08 (0.91 to 1.28) | 0.37 | | |
| Analgesics received | 1.28 (1.08 to 1.52) | **<0.001** | 1.33 (1.17 to 1.51) | **<0.001** |
| Increase in one type of polypharmacy | 1.01 (0.93 to 1.11) | 0.75 | | |
| Presence of other psychiatric conditions (within 1 year before or after diagnosis of mood (affective) disorders) | | | | |
| F00–F03 (dementia) | 1.14 (0.96 to 1.41) | 0.12 | | |
| F20–F29 (schizophrenia spectrum disorder) | 1.11 (0.91 to 1.38) | 0.3 | | |
| F30–F31 (bipolar affective disorder) | 0.81 (0.67 to 0.98) | **0.03** | 0.80 (0.68 to 0.95) | **0.01** |
| F40–F48 (neurotic to stress-related and somatoform disorders) | 0.82 (0.68 to 0.99) | **0.04** | 0.83 (0.70 to 0.99) | **0.04** |
| One unit increase in HoNOS | | | | |
| Cognitive problems | 1.02 (0.87 to 1.19) | 0.83 | | |
| Physical illness | 1.36 (1.19 to 1.56) | **<0.001** | 1.48 (1.31 to 1.67) | **<0.001** |
| Hallucinations | 0.91 (0.76 to 1.10) | 0.34 | | |
| Depressed mood | 0.83 (0.72 to 0.94) | **0.01** | 0.91 (0.82 to 1.00) | **0.05** |
| Relationship problems | 0.89 (0.77 to 1.02) | 0.1 | | |
| Daily living problems | 1.12 (0.98 to 1.29) | 0.11 | | |
| Occupational problems | 0.91 (0.79 to 1.04) | 0.15 | | |
| Admitted to general hospital (within 1 year before or after diagnosis of mood (affective) disorders) | | | | |
| Ischaemia+CHD+IHD | 1.01 (0.85 to 1.22) | 0.88 | | |
| Arrhythmia+AF | 1.03 (0.86 to 1.23) | 0.78 | | |
| Heart failure | 1.44 (1.12 to 1.86) | **0.01** | 1.43 (1.14 to 1.80) | **<0.001** |
| Diabetes | 1.22 (1.04 to 1.44) | **0.02** | 1.23 (1.06 to 1.44) | **<0.001** |
| Hypotension | 1.18 (0.93 to 1.48) | 0.17 | | |
| Hypercholesterolaemia | 0.85 (0.71 to 1.03) | 0.1 | | |

Continued

**Table 4** Continued

| Characteristics | Model 1 (n=20936) | | Model 2 (n=22322) | |
| --- | --- | --- | --- | --- |
| | HR (95% CI) | P value | HR (95% CI) | P value |
| Hypertension | 1.05 (0.90 to 1.23) | 0.53 | | |
| Urinary tract infections | 1.49 (1.28 to 1.74) | **<0.001** | 1.52 (1.32 to 1.75) | **<0.001** |
| Osteoporosis | 7.31 (5.82 to 9.18) | **<0.001** | 7.00 (5.65 to 8.70) | **<0.001** |
| Visual disturbance and blindness | 1.06 (0.74 to 1.53) | 0.74 | | |
| Hearing loss | 1.40 (1.03 to 1.92) | **0.03** | 1.46 (1.09 to 1.96) | **0.01** |
| Syncope or collapse | 1.18 (1.02 to 1.36) | **0.03** | | |
| Parkinson's disease | 0.85 (0.58 to 1.25) | 0.41 | | |
| Falls before mood (affective) disorder diagnosis | 1.02 (0.85 to 1.22) | 0.83 | | |
| Fractures before mood (affective) disorder diagnosis | 1.26 (1.01 to 1.57) | **0.04** | 1.33 (1.10 to 1.63) | **<0.001** |
| Increase in one attendance to A&E following mood (affective) disorder diagnosis | 1.01 (1.01 to 1.01) | **<0.001** | 1.01 (1.01 to 1.01) | **<0.001** |

Factors with a p value < 0.05 are marked in bold
A&E, accident and emergency services; AF, atrial fibrillation; CHD, coronary heart disease; HoNOS, Health of the Nation Outcome Scales; IHD, ischaemic heart disease; IMD, Index of Multiple Deprivation.

fractures, which could be due to increased fear avoidance and reduced activities due to stress-related disorders, leading to fewer falls.[42] We also found the presence of bipolar affective disorder and manic episodes to be protective factors against increased fracture risk. Although there has been little evidence investigating the role of bipolar affective disorder or its characteristics (including manic symptoms) in the risk of fracture, preliminary evidence has suggested an association between the use of lithium (ie, a primary treatment for bipolar disorder) and decreased fracture risk,[43] the preservation and enhancement of bone mass.[44]

Surprisingly, neither antidepressants nor antipsychotics were associated with risks of falls or fracture, despite documented associations between antipsychotic medications and reduced bone metabolism[45] and BMD.[46] Evidence on the association between antidepressants and BMD is ambiguous: while some authors reported that some antidepressants, especially selective serotonin reuptake inhibitors (SSRIs), have a negative impact on bone strength,[47 48] other findings have been null.[49] However, we did not attempt to account for specific types of antidepressants in the analysis, and previous studies indicate that tricyclic antidepressants have no adverse effect on bone health.[28] On the other hand, analgesic use was associated with increased risks of falls and fractures, consistently with previous findings,[50] and possibly mediated by central nervous system effects, such as dizziness.[51] Antihypertensive drug use was associated with a decreased fracture risk, confirming previous results,[52] potentially due to their improving effect on calcium absorption.[53]

Finally, extensive evidence has indicated several physical conditions as risk factors for falls or fractures in the general population,[54 55] including UTI[56] and diabetes.[57] It is known that mental health disorders (eg, depression) increase the chance of major physical comorbidities.[58] However, few studies have established the impact of comorbid physical conditions on risks of falls and fractures among people with affective disorders. Our study indicates heart failure, diabetes, UTI and osteoporosis to be associated with hospitalisation due to falls or fractures. Hypotension and syncope or collapse were additionally associated with an increased risk of falls, and hearing loss was associated with an elevated fracture risk. Unsurprisingly, osteoporosis was strongly associated with falls and fractures, as it has been acknowledged as the most important yet potentially treatable factor for falls[54] and fractures[59] due to its effect on BMD. Moreover, osteoporosis tends to occur simultaneously with heart failure[60] due to shared risk factors such as diabetes. Unsurprisingly, syncope or collapse was also strongly associated with an increased risk of falls, as syncope is frequently mistaken for falls, to the point that the European Society of Cardiology has emphasised the need to explore syncope as an independent factor leading to falls.[61] Diabetes was another risk factor for both falls and fracture, confirming the association found in the general population.[62] Hearing loss was the only factor associated with an elevated risk of fracture in our study, an association previously confirmed in the general population.[63] This finding may be explained by the damaging impact of hearing loss on orientation skills and physical activity.[64] Finally, our study confirmed

the association between history of falls and increased risk of future falls, previously reported in the general population[34 65] and in people with depression.[66] Moreover, our data confirmed how a prior fracture represents a 50%–100% higher risk of experiencing future fracture, as reported in the general population.[67]

The main strengths of our study are: (1) analyses controlling for various confounders; and (2) a large sample including people with clinically diagnosed affective disorders. However, there are important limitations. First, we did not stratify according to different affective disorder subtypes which could report different risk factors; similarly, we analysed psychotropic medication categories but did not investigate specific agents or subcategories. Second, we had no information on the types of fractures reported at admission and on factors (eg, balance) that are influenced by neuropathological lesions in people with depression.[68] Additionally, the majority of studies investigating falls and fractures rely on self-report data while our study relied on ICD hospitalisation codes increasing reliability. However, the causes of fractures and the types of falls could not be explored in the current study, although we investigated factors associated with falls across the whole sample from the rich mental health record database. Future research should attempt to disentangle of the potential risk factors/causes of falls and fractures research in affective disorders. A limitation of health administrative data is some inconsistency or limited data on the granular detail. Third, lifestyle factors, such as alcohol consumption,[31 38] could not be explored. Fourth, our study only ascertained information on physical comorbidities from hospitalisation records and could have overlooked patients with mild osteoporosis. Additionally, physical comorbidities were ascertained within 6 months of diagnosis, and patients who were hospitalised for a fall or fracture within these 6 months would be more likely to have their osteoporosis ascertained. Fifth, falls and fractures are closely related and often co-occur.[69] In this study, although falls and fractures were identified from ICD hospitalisation codes, falls and fractures categories were not mutually exclusive, as a patient could be counted in both the fall and fracture cohort if he/she had a fall and a fracture in the same event. Moreover, factors such as inaccurate recording and fine distinction between ICD hospitalisation codes for falls and fractures could also potentially hinder our attempt to differentiate the two concepts. Thus, further work is needed to elucidate the potential overlap and distinct relationship among falls and fractures in people with affective disorders. Finally, since medication records were ascertained within 1 year of diagnosis, we cannot assess whether medications were prescribed as a result or as a cause of falls for those who had a fall or fracture within 6 months of their diagnosis.

## CONCLUSION

Our study reports that over a mean 5 years' follow-up, approximately 8% of a large cohort with affective disorders was hospitalised due to either a fall or fracture. Factors such as older age, certain medications and having a history of fall or fracture are significant predictors, broadly comparable with risk factors established in the general population. Heavy hospital burden following a fall or fracture was also described.

Our current study provides important implications for future research and clinical practice. Future research should take into account the limitations listed above and potentially explore the mechanisms through which the factors identified in this study (eg, physical conditions such as UTI, or medication use) may impact risks of falls and fractures in people with affective disorders. The associations between certain factors (eg, psychotropic medication use) and risk of falls and fractures were not confirmed in the current study, thus future studies are encouraged to replicate our findings and explore these factors further.

In terms of implications for future clinical practice, given the lack of osteoporosis screening for people with mental health problems in current clinical settings,[70] BMD checks and osteoporosis screening should be a routine for people with affective disorders, especially if older and with established comorbid chronic illnesses. Fall assessments (eg, the Fracture Risk Assessment Tool) should be routinely conducted for people with affective disorders, since approximately 70% of low-energy fractures are due to falls.[71] Fall/fracture prevention programmes should also be offered due to an average 14% reduction in falls risk in the general population following fall prevention programmes.[72] For example, the combination of cognitive–behavioural therapy and exercise has been previously found to be effective in improving depressive symptoms[73] and self-efficacy[74] in the general population. The effectiveness of these prevention programmes should also be examined in people with affective disorders and subsequently provided if their benefits are confirmed.

**Author affiliations**
[1]Institute of Psychiatry, Psychology and Neuroscience (IoPPN), King's College London, London, UK
[2]Department of Rehabilitation Sciences, KU Leuven - University of Leuven, Leuven, Belgium
[3]University Psychiatric Centre KU Leuven, KU Leuven - University of Leuven, Leuven, Belgium
[4]Research and Development Unit, Parc Sanitari Sant Joan de Deu, Sant Boi de Llobregat, Spain
[5]Institució Catalana de Recerca i Estudis Avancats (ICREA), Barcelona, Spain
[6]South London and Maudsley NHS Foundation Trust, Institute of Psychiatry, London, UK

**Contributors** BS acquired funding for the study and the guarantor of this research. GP conducted the analysis with support from all coauthors. RM drafted introduction and discussion sections, ER drafted results section and all authors (BS, GP, DV, AK, RS, CM) provided critical revisions and approved the final version. The PPI group helped conceptualise and co-develop the study, as well as contributed to the funding applications.

**Funding** BS is supported by a Clinical Lectureship (ICA-CL-2017-03-001) jointly funded by Health Education England (HEE) and the National Institute for Health Research (NIHR). BS is also supported by an NIHR Advanced fellowship

(NIHR301206, 2021-2026). Brendon Stubbs is on the Editorial board of Ageing Research Reviews, Mental Health and Physical Activity, The Journal of Evidence Based Medicine and The Brazilian Journal of Psychiatry. Brendon has received honorarium from a co-edited a book on exercise and mental illness and advisory work from ASICS for unrelated work. RM and ER were supported by a grant from Guy's & St Thomas' Charity. RS receive salary support from the National Institute for Health Research (NIHR) Biomedical Research Centre (BRC) at South London and Maudsley NHS Foundation Trust and King's College London, and RS is a NIHR Senior Investigator and is funded by the National Institute for Health Research (NIHR) Applied Research Collaboration South London (NIHR ARC South London) at King's College Hospital NHS Foundation Trust.

**Disclaimer** The views expressed are those of the author(s) and not necessarily those of mentioned above.

**Competing interests** None declared.

**Patient consent for publication** Not required.

**Ethics approval** This study has full approval for secondary analysis of CRIS (Oxford Research Ethics Committee C, reference 18/SC/0372).

**Provenance and peer review** Not commissioned; externally peer reviewed.

**Data availability statement** Data are available upon reasonable request. Data are available on reasonable request. For questions regarding the study, please contact RM at ruimin.1.ma@kcl.ac.uk.

**Author note** CM and BS are joint last authors.

**ORCID iDs**
Ruimin Ma http://orcid.org/0000-0002-6051-8093
Gayan Perera http://orcid.org/0000-0002-3414-303X
Ai Koyanagi http://orcid.org/0000-0002-9565-5004

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
