## [Reviewer comments · BMJ Open]

ARTICLE DETAILS

TITLE (PROVISIONAL)	Predictors of falls and fractures leading to hospitalisation in 36,101 people with affective disorders: A large representative cohort study
AUTHORS	Ma, Ruimin; Perera, Gayan; Romano, Eugenia; Vancampfort, Davy; Koyanagi, Ai; Stewart, Robert; Mueller, Christoph; Stubbs, B

VERSION 1 – REVIEW

REVIEWER	Majeed, Haroon Manchester University NHS Foundation Trust, Department of Trauma and Orthopaedic Surgery
REVIEW RETURNED	15-Aug-2021

GENERAL COMMENTS	This article is an excellent effort by the authors on two important clinical areas; mental health disorders and the fractures. It is well-conducted, well-structured, and well-presented with appropriate details to cover the relevant aspects. In my view, this will be a valuable addition to the existing literature on this subject and will be interesting for the readers. I have nothing to suggest to improve it, except to standardise the spellings of 'hospitalisation' either to British English format or the American as currently its written in both formats in different places.
--

REVIEWER	Adamczewska, Natalia Bournemouth University
REVIEW RETURNED	28-Nov-2021

GENERAL COMMENTS	The paper addresses an important issue of falls and fractures among individuals with affective disorders, thus this large study is of much value. However, the main issue related to the subject of the study is the problem of distinction of falls and fractures occurrences. There is much overlap between the two, since some falls result in fractures and some fractures are caused by falls. There is a need for some clarification of how those two were differentiated. It is particularly important since the result section mentions the occurrence of falls with fractures. The authors acknowledge that they had no information on type of fractures, however, another limitation is a lack of information on the causes of fractures, as well as no information on a type of falls, such as low falls, high falls, motor vehicle crashes, assault, sports injuries, and so on, which might be particularly relevant in the area of affective disorders. Table 1 includes age at the time of mood disorders diagnosis, however, the paper does not seem to mention the age of
---

	fall/fracture occurrence. In fact, it appears that age was not an inclusion/exclusion criterion, thus that ought to be stated in the paper. Presumably, a list of inclusion and exclusion criteria would benefit the paper.
--	---

REVIEWER	Campani, Daiana Università del Piemonte Orientale
-----------------	--

REVIEW RETURNED	11-Jan-2022
-------------

GENERAL COMMENTS	Thank you for the opportunity to review this paper titled, "Predictors of falls and fractures leading to hospitalization in 36,101 people with affective disorders: A large representative cohort study". This article is regarding fall predictors which is an important area of concern as our population ages. Following revisions were recommended; Introduction: Rationale of the study and expected benefits should be written more clearly in the last paragraph of the introduction. Conclusion: Should be written more comprehensively. Relevancy, recommendations and implications for further researches could be stronger. Thank you for the opportunity to review this interesting article.
--

VERSION 1 – AUTHOR RESPONSE

Reviewer: 1

Dr. Haroon Majeed, Manchester University NHS Foundation Trust

Comments to the Author:

This article is an excellent effort by the authors on two important clinical areas; mental health disorders and the fractures. It is well-conducted, well-structured, and well-presented with appropriate details to cover the relevant aspects. In my view, this will be a valuable addition to the existing literature on this subject and will be interesting for the readers. I have nothing to suggest to improve it, except to standardise the spellings of 'hospitalisation' either to British English format or the American as currently its written in both formats in different places.

Reply: We would like to express our sincerely gratitude to the reviewer for acknowledging the importance of our work.

We agree with the reviewer's comment, we have checked the whole manuscript, 'hospitalization' has been changed to 'hospitalisation' to keep it consistent.

Reviewer: 2

Dr. Natalia Adamczewska, Bournemouth University

Comments to the Author:

The paper addresses an important issue of falls and fractures among individuals with affective disorders, thus this large study is of much value. However, the main issue related to the subject of the study is the problem of distinction of falls and fractures occurrences. There is much overlap between the two, since some falls result in fractures and some fractures are caused by falls. There is a need for some clarification of how those two were differentiated. It is particularly important since the result section mentions the occurrence of falls with fractures.

Reply: We would like to thank the reviewer for acknowledging the importance of work and offering valuable comments which have helped improve the quality of our manuscript.

We agree with the reviewer's point on the overlap between falls and fractures. For the current study, we obtained falls and fracture codes from ICD-10 diagnosis, mentioned anywhere from primary diagnosis to any secondary diagnoses. Thus, our falls and fractures categories are not mutually exclusive. For example, in the records, if the patient had a fall and a fracture in the same event, we would have counted them in the falls cohort and in the fractures cohort.

Therefore, to acknowledge the overlap between falls and fractures, we added the following sentences in the introduction (page 4):

'They often overlap, with more than 90% of hip fractures caused by falls [3] and share similar and multifactorial causes..'

'However, management is difficult, as causes for falls and fractures can be both intrinsic and extrinsic, and prevention requires both pharmacologic and nonpharmacologic interventions [3].'

We also added the following sentences in the limitation section to acknowledge this limitation of our research (page 17):

'Fifth, falls and fractures are closely related and often co-occur [69]. In this study, although falls and fractures were identified from ICD hospitalisation codes, falls and fractures categories were not mutually exclusive, as a patient could be counted in both the fall and fracture cohort if he/she had a fall and a fracture in the same event. Moreover, factors such as inaccurate recording and fine distinction between ICD hospitalisation codes for falls and fractures, could also potentially hinder our attempt to differentiate the two concepts. Thus, further work is needed to elucidate the potential overlap and distinct relationship among falls and fractures in people with affective disorders.'

The authors acknowledge that they had no information on type of fractures, however, another limitation is a lack of information on the causes of fractures, as well as no information on a type of falls, such as low falls, high falls, motor vehicle crashes, assault, sports injuries, and so on, which might be particularly relevant in the area of affective disorders.

Reply: We thank the reviewer for this helpful comment. The limitation on not including causes of fractures and types of fractures has been acknowledged in our limitation section (Page 17). The following sentences have been added on page 17:

'Additionally, the majority of studies investigating falls and fractures rely upon self-report data whilst our study relied on ICD hospitalisation codes increasing reliability. However, the causes of fractures and the types of falls could not be explored in the current study, although we investigate factors associated with falls across the whole sample from the rich mental health record database. Future

research should attempt to disentangle of the potential risk factors/causes of falls and fractures research in affective disorders. A limitation of health administrative data is some inconsistency or limited data on the granular detail. ‘

Table 1 includes age at the time of mood disorders diagnosis, however, the paper does not seem to mention the age of fall/fracture occurrence. In fact, it appears that age was not an inclusion/exclusion criterion, thus that ought to be stated in the paper. Presumably, a list of inclusion and exclusion criteria would benefit the paper.

Reply: We really appreciate the reviewer’s comment on the age of fall/fracture occurrence.

In the method section (page 6), we mentioned that ‘All SLAM patients with mood [affective] disorders (ICD10 codes F30-F34), diagnosed between 1st of January 2008 and 31st March 2016, were included’. This is the inclusion criteria of the current study.

Thank you, the reviewer, for pointing out the lack of exclusion criteria, the following sentences have been added following the inclusion criteria (page 6):

‘Those patients with an age under 18 at the time of mood disorders were excluded.’

Regarding age at time of mood disorder diagnosis, and age at falls and fracture occurrence, please see the table below. The table shows mean age at falls/ fractures at the time of Mood [affective] disorders diagnosis and at the time of fall or fracture those who had a fall/ fracture for our sample. Our sample had a fall 1.9 years after mood disorder diagnosis and had a fracture roughly the same period, 1.9 years after mood disorder diagnosis.

	Presence of falls	Presence of fractures
Age	No (n= 34,270) Yes (n= 1,831)	No (n= 33,908) Yes (n= 2,193)
Mean age at diagnosis (SD)	43.4 (17.1)	62.4 (19.9)
Mean age at fall/ fracture/ (SD)	43.3 (17.1)	60.6 (20.6)
	64.2 (20.8)	62.5 (22.8)

The following sentence has been added to the result section (page 10):

‘(mean age at first fall 64.2, SD: 24.8; mean age at first fracture 62.5, SD: 22.8 (Supplementary material Table 1).’

The table has also been added in the manuscript as a supplementary material.

Once again, we thank expert reviewer for improving our manuscript.

Reviewer: 3

Dr. Daiana Campani, Università del Piemonte Orientale

Comments to the Author:

Thank you for the opportunity to review this paper titled, "Predictors of falls and fractures leading to hospitalization in 36,101 people with affective disorders: A large representative cohort study". This article is regarding fall predictors which is an important area of concern as our population ages.

Reply: We want to thank the reviewer for acknowledging the importance of our work.

Following revisions were recommended;

Introduction:

Rationale of the study and expected benefits should be written more clearly in the last paragraph of the introduction.

Reply: We really appreciate the reviewer's suggestions. The last paragraph of the introduction has been amended to clarify the rationale and expected benefits of our research (page 4):

'Given the increased concerns about the multifactorial causes of falls and fractures [3, 4, 5], and as an answer to the lack of large-scale studies on the matter, this study examines a large representative cohort study to investigate predictors of falls and fractures leading to hospitalisation among people with clinically diagnosed affective disorders. In this way, we aim to provide results based on clinical diagnosis rather than self-reported data [14, 15], presenting a solid overview on potential risk factors for falls and fractures in the high-risk population of people with affective disorders.'

Conclusion:

Should be written more comprehensively. Relevancy, recommendations and implications for further researches could be stronger.

Thank you for the opportunity to review this interesting article.

Reply: We agree with the reviewer's suggestion and want to thank the reviewer for further improved our manuscript. In light of this, we have updated the conclusions on implications of future research and clinical practice have been updated (page 18):

'Our current study provides important implications for future research and clinical practice. Future research should take into account the limitations listed above and potentially explore the mechanisms through which the factors identified in this study (e.g., physical conditions such as UTI, or medication use) may impact risks for falls and fractures in people with affective disorders. The associations between certain factors (e.g., psychotropic medication use) and risk for falls and fractures were not confirmed in the current study, future studies are encouraged to replicate our findings and explore these factors further.

In terms of implications for future clinical practice, given the lack of osteoporosis screening for people with mental health problems in current clinical settings [70], BMD checks and osteoporosis screenings should be a routine for people with affective disorders, especially if older and with established co-morbid chronic illnesses. Fall assessments (e.g., the Fracture Risk Assessment Tool) should be routinely conducted for people with affective disorders, since approximately 70% of low-energy fractures are due to falls [71]. Fall/fracture prevention programmes should also be offered due to an average 14% reduction in falls risk in the general population following fall prevention programmes [72]. For example, the combination of cognitive behavioural therapy and exercise have been previously found to be effective in improving depressive symptoms [73] and self-efficacy [74] in the general population. The effectiveness of these prevention programmes should also be examined in people with affective disorders and subsequently provided if their benefits are confirmed.'